# Non-Invasive Mapping of Cerebral Autoregulation Using Near-Infrared Spectroscopy: A Study Protocol

**DOI:** 10.3390/mps6030058

**Published:** 2023-06-09

**Authors:** Amanjyot Singh Sainbhi, Nuray Vakitbilir, Alwyn Gomez, Kevin Y. Stein, Logan Froese, Frederick A. Zeiler

**Affiliations:** 1Department of Biomedical Engineering, Price Faculty of Engineering, University of Manitoba, Winnipeg, MB R3T 5V6, Canada; vakitbir@myumanitoba.ca (N.V.); steink34@myumanitoba.ca (K.Y.S.); froesel3@myumanitoba.ca (L.F.); frederick.zeiler@umanitoba.ca (F.A.Z.); 2Section of Neurosurgery, Department of Surgery, Rady Faculty of Health Sciences, University of Manitoba, Winnipeg, MB R3A 1R9, Canada; gomeza35@myumanitoba.ca; 3Department of Human Anatomy and Cell Science, Rady Faculty of Health Sciences, University of Manitoba, Winnipeg, MB R3E 0J9, Canada; 4Centre on Aging, University of Manitoba, Winnipeg, MB R3T 2N2, Canada; 5Division of Anaesthesia, Department of Medicine, Addenbrooke’s Hospital, University of Cambridge, Cambridge CB2 0QQ, UK; 6Department of Clinical Neuroscience, Karolinska Institutet, 171 77 Stockholm, Sweden

**Keywords:** cerebrovascular reactivity system, cerebral autoregulation, cerebral heat maps, high spatial resolution, healthy volunteers, high temporal resolution, near infrared spectroscopy, neuroimaging system, NIRS-based indices, non-invasive system

## Abstract

The ability of cerebral vessels to maintain a fairly constant cerebral blood flow is referred to as cerebral autoregulation (CA). Using near-infrared spectroscopy (NIRS) paired with arterial blood pressure (ABP) monitoring, continuous CA can be assessed non-invasively. Recent advances in NIRS technology can help improve the understanding of continuously assessed CA in humans with high spatial and temporal resolutions. We describe a study protocol for creating a new wearable and portable imaging system that derives CA maps of the entire brain with high sampling rates at each point. The first objective is to evaluate the CA mapping system’s performance during various perturbations using a block-trial design in 50 healthy volunteers. The second objective is to explore the impact of age and sex on regional disparities in CA using static recording and perturbation testing in 200 healthy volunteers. Using entirely non-invasive NIRS and ABP systems, we hope to prove the feasibility of deriving CA maps of the entire brain with high spatial and temporal resolutions. The development of this imaging system could potentially revolutionize the way we monitor brain physiology in humans since it would allow for an entirely non-invasive continuous assessment of regional differences in CA and improve our understanding of the impact of the aging process on cerebral vessel function.

## 1. Introduction

Cerebral autoregulation (CA) is the innate ability of the cerebral vessels to maintain a relatively constant cerebral blood flow (CBF) over a range of systemic arterial pressures [1,2]. Cerebrovascular reactivity (CVR) is the mechanism behind the maintenance of constant CBF through the constriction and dilation of cerebral blood vessels [1,2]. CA tends to remain fairly constant between the lower limit of autoregulation (LLA) and upper limit of autoregulation (ULA) on the Lassen autoregulatory curve [3]. It is important to mention that CVR is a broader term for CA to describe the physiologic process; therefore, CVR and CA are not entirely interchangeable since CVR can occur outside of the limits of autoregulation [4]. Nevertheless, for a metric to be considered a CA metric, it needs to correctly detect the LLA and ULA. Currently, NIRS metrics are only able to assess the LLA in preclinical literature, which is the reason we use CVR hereafter in maintaining a consistent nomenclature for metrics assessing CA that have not been fully validated against both the LLA and ULA. Currently, there is a lack of understanding of the biological impacts of age and sex on CA, with impaired delivery of blood flow and nutrients to the brain leading to ongoing secondary brain damage. This secondary damage can lead to strokes with deficient blood flow starving the brain tissue, or lead to swelling and hemorrhage due to excessive blood flow [5,6,7,8,9]. Our limited ability to continuously assess cerebral blood vessel function in humans and characterize its regional disparity in both high temporal and spatial resolutions is a large contributing factor to the existing knowledge gap regarding the incidence and consequences of impaired CA. Preliminary attempts to characterize CA, to date, have relied on intermittent “snap shots” of CBF using advanced neuroimaging with perfusion of functional sequencing [10], or on very focal assessments using biomedical engineering signal processing of data obtained from invasive/non-invasive cerebral monitoring devices [11,12]. The main limitation of these preliminary works, leading to the current knowledge gap, is both the poor temporal resolution of advanced neuroimaging studies and the inability to monitor multiple brain regions with current bedside continuous techniques. It becomes clear, based on the abovementioned facts, that our ability to characterize cerebrovascular function accurately and continuously in humans is insufficient.

An invasive, intracranial pressure (ICP)-derived metric, pressure reactivity index (PRx), is the most established method for CA assessment [13,14]; however, this is limited by the spatial resolution and requires neurosurgical/neurocritical care expertise due to the invasive ICP monitoring. Current clinical care has been focusing on spontaneous CA assessments with PRx as a prime example [7,15,16,17]. A potential method for the derivation of continuous CA indices in humans is near-infrared spectroscopy (NIRS) [14,15,18,19]. This technique uses continuous NIRS-based oxy-/deoxy-hemoglobin (HbO/HHb) or regional tissue oxygen saturation (rSO_2_) measures as a surrogate for pulsatile cerebral blood volume (CBV) [12,19,20,21,22,23], where rSO_2_ might also be a measure for changes in flow [18]. Paired with arterial blood pressure (ABP) monitoring, it is able to assess continuous CA through the temporal relationship between slow-wave fluctuations in a surrogate measure of pulsatile CBV (i.e., NIRS measures) with a driver for flow (i.e., ABP) using sliding Pearson correlation calculations to derive an index ranging from −1.0 to 1.0 [12,15,18,20]. Validating evidence supporting these NIRS-based CA metrics as measures of the lower limit of CA have been provided in experimental animal literature [14,24,25] and this approach has been described as a substitute for ICP-based indices [14,15,26]. Recent work has described the technique of NIRS-derived CA measures using commercially available NIRS devices (INVOS 5100C) and non-invasive ABP assessments (Finapres NOVA) [20,21,22]. The current commercial NIRS systems have approval to be used in clinical setting but they lack multi-channel functionality since they are typically limited to two channels (i.e., bifrontal assessments) with a poor sampling frequency of ~1 Hz [17,20,21,22] and they only measure NIRS parameters such as rSO_2_, HbO or HHb. Currently, there is no FDA- or Health-Canada-approved device that measures CVR using a NIRS device. The need for multi-channel functionality in NIRS is to be able to assess cerebral regions, other than the frontal region, using CVR metrics to properly assess each region of the brain, which cannot be estimated or derived by only measuring the frontal region. This has clinical utility in pathologic states that do not uniformly impact cerebrovascular physiology such as traumatic brain injury (TBI) or delayed cerebral ischemia following aneurysmal subarachnoid hemorrhage (SAH).

Thus, to improve the understanding of continuously assessed CA in humans, we need a tool with both improved spatial and temporal resolution. With recent advances in NIRS technology, custom multi-channel NIRS systems are available that offer sampling frequencies up to 250 Hz, enabling the simultaneous data acquisition of full cerebral pulse waveform from multiple brain regions [27,28,29,30]. Since these are research systems, they cannot be used in a clinical setting to direct care provision until they acquire the appropriate health safety approvals. Non-overlapping moving average filters are used to deal with the data overload issue with high-resolution CVR data [31,32,33] since the large volumes of data are most likely beyond what a human can process (and these have been suggested as a contributor to burnouts [34,35,36,37,38,39]). Bridging this existing technological gap in high spatial and temporal resolution CA monitoring system, we have recently proposed to employ an advanced multi-channel NIRS system coupled with an entirely non-invasive continuous arterial blood pressure (niABP) monitor to create a new wearable and portable imaging system that derives CA maps of the entire brain with high sampling rates at each point. Recent work from our lab has demonstrated the feasibility of developing such a system, with a functional prototype currently running (Figure 1) [30].

Thus, to further advance our recent technological development, there exists a need for additional optimization/testing in a human population and preliminary exploration of normative data values for such CA maps in a large healthy volunteer population, accounting for age and biological sex. We outline the approved protocol for such a study and its specific objectives/hypotheses related to testing/optimizing the new platform and collecting a large healthy volunteer population to preliminarily evaluate the impact of chronological age and biological sex on CA.

## 2. Experimental Design

We proposed a prospective observational study, split into two phases to address both optimization/testing of the developed platform and characterization of a normative dataset for CA maps in a healthy population. This will be accomplished through two main objectives:

### 2.1. Objective 1

To perform in vivo testing and optimization of the device using a block-trial design to evaluate the CA mapping system’s performance during various perturbations. To assess functionality and feasibility of real-time application in humans, in vivo testing will occur on a small population of healthy human volunteers. Such work will investigate the feasibility of the combined monitoring setup, optimal NIRS channel placement using the adjustable OxyMon NIRS cap (shown in Figure 2), ability of pipelines to extract and analyze signals in real time for continuous derivation of NIRS-based CVR metrics at each channel, and functionality of CVR map generation in real-time. Finally, we will investigate the perturbations to the system through a series of testing (block design separated by baseline rest): A. transient hyperemic response testing via carotid compression methods [10,41]; B. orthostatic challenge responses (lying-to-sit, sit-to-stand) [10]; C. vascular chemo-reactivity via fast and slow breathing exercises [42]; and D. impact of neurovascular coupling through cognitive/Stroop testing [43,44,45,46]. The work carried out here will establish areas for improvement in real-time. The testing and optimization phase will last 18 months.

 **Hypothesis 1:**
*The CA mapping system will display good variation and responsiveness in signals to: A. transient hyperemic testing through carotid compression; B. orthostatic challenge; C. chemo-reactivity to manipulations in end-tidal carbon dioxide (CO_2_) (EtCO_2_); and D. neurovascular coupling. CA map generation and point/interval predictions will be possible in real-time using this system during the block-trial testing.*


### 2.2. Objective 2

To explore the impact of aging and sex on regional disparities in CA in a healthy volunteer population using static recording along with perturbation testing. The current understanding of the aging process and sex on continuously assessed CA, and regional disparities, has been hampered by a lack of a high-resolution platform. We will evaluate the impact of both age and sex on CA using the newly developed platform through static CA assessments in 200 healthy volunteers. Within the 1 h period, volunteers will have their age and sex recorded in a de-identified electronic database, have an hour of non-invasive simultaneous 24-channel OxyMon NIRS and Finapres NOVA ABP recording, which is divided into static resting state recording and perturbation testing. This will facilitate the derivation of a normative reference range for CA using the platform, in addition to making the dataset richer. Static resting state assessments will be carried out over the first half hour, and in the second half hour, perturbations to the system will be investigated through a series of testing separated by baseline rest: A. orthostatic challenge responses (lying-to-sit, sit-to-stand) [10]; B. vascular chemo-reactivity via fast and slow breathing exercises [42]; and C. impact of neurovascular coupling through Stroop testing [43,44,45,46].

 **Hypothesis 2:**
*Such work will involve the use of the designed platform to explore the aging process and sex discrepancies in continuously assessed CA using the novel system statically along with responsiveness in signals to: A. orthostatic challenge; B. chemo-reactivity to manipulations in EtCO_2_; and C. neurovascular coupling to Stroop testing.*


### 2.3. Recruitment of Human Participants

Studies for both objectives will be conducted at the Winnipeg Acute TBI Labs, located in a non-clinical area of Health Sciences Center (HSC) in Winnipeg, Manitoba, Canada. The volunteers will be healthy volunteers with no history of colour-blindness, stroke, brain disorder, neurological illness, pulmonary disease, systemic vascular or cerebrovascular disease, and will be screened for this prior to enrollment in either objective. For Objective 1, we plan on recruiting 50 healthy volunteers which is based on the minimum number of volunteer for CA time-domain studies established at 40 [47]. For Objective 2, we plan on recruiting 200 healthy volunteers, exceeding the minimum sample size needed for such physiologic work [47], with age ranging from 18 to 90+ years and at least 20 individuals per decade of age with a cap of 40 individuals. To avoid a huge imbalance in age, we will audit the age/sex distributions as we go along to avoid over-recruitment in a specific age bracket. We have a cap of 40 individuals for a certain age decade to avoid age imbalances. There will be an equal representation of males and females to enable the assessment of sex-related differences. The only patient demographic information to be collected is age and biological sex designation.

### 2.4. Physiologic Monitoring Systems

Multi-channel NIRS signals will be recorded using the functional NIRS system OxyMon Mk III (Artinis Medical Systems, Elst, The Netherlands), shown in Figure 3, which is a non-invasive NIRS research system not under the purview of Health Canada as it is not considered a medical device. The functional NIRS system is equipped with eight transmitting optodes and eight receiving optodes, along with additional eight reference optodes. The combination of a transmitter and receiver optode pair creates normal channels separated by a distance of 30 mm, whereas the combination of a transmitter and reference optode pair creates a short channel separated by a distance of 10 mm. The short channel will be subtracted from the normal channel to eliminate scalp noise, and it is important to note that this system is able to eliminate scalp noise separately at each channel. The continuous niABP signal will be obtained using Finapres Nova (Finapres Medical Systems, Enschede, The Netherlands), which uses the finger-cuff technique and this niABP system has been approved by Health Canada. Additionally, a regional cerebral oxygen saturation (rSO_2_) signal will be obtained using INVOS 5100C or 7100 (Medtronic of Canada Ltd., Brampton, ON, Canada) acquired with a single channel over a frontal lobe and this commercial NIRS system has been approved by Health Canada. This device provides a “gold standard” signal source for ground-truth comparison.

### 2.5. High-Frequency Data Acquisition

Various signals will be obtained through non-invasive methods (NIRS and niABP), with all signals recorded in high-frequency time series using ICM+ software (Cambridge Enterprise Ltd., Cambridge, UK) connected to the data streams from the two systems. Signals from all the monitoring devices described below will be recorded in time series using this software throughout the recording period for each objective.

ABP will be obtained with the Finapres device that can output the full waveform at up to 250 Hz sampling frequency, sampled through an entirely non-invasive finger-cuff. Signals from the NIRS system at each channel would be HbO, HHb, total hemoglobin (tHb) and the difference between HbO, and HHb (HbDiff) of regular channels, reference channels, and the difference reference from regular channels, in addition to recording raw optical densities (OD) values. These NIRS signals will be recorded using a total of eight channels with one channel per brain lobe (frontal, parietal, occipital and temporal lobes) resulting in four channels being placed on the left and right hemispheres of the brain each. In addition to the eight normal channels, the signals from eight reference channels will also be recorded, and the NIRS system is able to output these signals at a sampling frequency of up to 250 Hz. Figure 4 shows an example of the NIRS and ABP signals. Additionally, a single rSO_2_ channel along with two OD signals, from the two detectors that are on a single sensor, will be obtained from the commercial NIRS system sampled at ~1 Hz. EtCO_2_ will be obtained using Capnostream 35 Portable Respiratory Monitor (Medtronic of Canada Ltd., Medtronic, Ottawa, ON, Canada) through a nose clip, sampled at 20 samples/second, and will be captured using an external storage device and subsequently linked via data/time stamp to other physiologic data streams.

The backend communications of these devices have been accomplished in our lab, facilitating success. The NIRS system uses its own software called OxySoft to perform and view measurements with an option to output the data in real time using Lab Streaming Layer (LSL; https://github.com/sccn/labstreaminglayer, accessed on 14 April 2023), which is a system for the unified collection of measurement time series, handling both networking and time-synchronization. The Finapres device outputs the ABP signal in an analogue format, which is digitized using Data Translations DT9804/DT9826 converters (Data Translation, Marlboro, MA, USA), and an LSL data streams is created using LSL. The commercial NIRS system outputs the signal using digital data transfer. To record these data streams from these systems using ICM+, custom modules in Python have been developed. These modules read the custom NIRS, commercial NIRS, and Finapres streams in LSL and output them to virtual communication (COM) ports to be recorded in ICM+ along with saving the signals in a Hierarchical Data Format version 5 (HDF5) file format.

### 2.6. Physiologic Data Processing

At each channel, four indices will be derived by first decimating the raw signals using non-overlapping moving average filters with a duration of 10 s, which will allow us to focus on the slow-wave vasogenic fluctuations associated with CA. Then, Pearson correlation coefficients will be calculated using 30 consecutive 10 s means from the NIRS and the ABP signals, which are updated every 10 s. These four indices are oxyhemoglobin index (HbOx—correlation between HbO and ABP), deoxyhemoglobin index (HHbx—correlation between HHb and ABP), total hemoglobin index (tHbx—correlation between tHb and ABP), and hemoglobin difference index (HbDiffx—correlation between HbDiff and ABP, which is the difference between HbO and HHb as given in Table 1).

The single channel NIRS signal will be recorded using commercial NIRS system INVOS 5100C or 7100. The calculation of Pearson correlation coefficients from the NIRS rSO_2_ signal and the ABP signal will produce a cerebral oxygen index (COx—correlation between rSO_2_ and ABP) for the single frontal channel. These data will be used as a “gold standard” to compare the larger higher-frequency NIRS system results.

### 2.7. General Statistical Methodologies

Statistical analysis will be performed utilizing R statistical software (R Foundation for Statistical Computing) and Python scripting language (Python 3, Scotts Valley, CA, USA: CreateSpace). To compare between patient age groups, sex, and regional differences, various techniques will be employed, and comparisons between various age ranges will also occur. Basic descriptive statistics will be performed, with the comparison between participant groups and variables being conducted via *t*-test, Mann–Whitney U, chi-square, analysis of variance (ANOVA), Kruskal–Wallis, Friedman and Jonckheere–Terpstra testing [48,49], where appropriate. General correlations will be described using Pearson/Spearman coefficients, where applicable. Univariate and multi-variate linear and non-linear modelling techniques will be explored between age, sex, and the various CA indices. Multi-variate testing for inter-index covariance between the NIRS-derived CA indices will be performed and compared between regions, sex groups, and age ranges. The following multi-variate techniques will be performed: principal component analysis (PCS), agglomerative hierarchal clustering (AHC), and k-means cluster analysis (KMCA) [50,51]. Time series methodologies will be used to explore the relationship between various CA metrics and raw recorded physiology using univariate and multi-variate autoregressive integrative moving average techniques, generating forecasting models stratified by age, to predict CA over varied intervals using Granger causality, impulse response function testing and vector ARIMA modelling. A detailed description of these techniques can be found in the referenced literature [51,52,53,54]. The advanced techniques described have been employed in our prior work [23,33,55,56,57,58]. Additionally, slow-wave power will be calculated for ABP and fNIRS signals using power spectral density analysis (PSD) [59] along with the analysis of the relationship between the slow-wave power of the signals and the CA capacity in both objectives.

### 2.8. Pitfall Mitigation

The advanced NIRS systems proposed carry limitations. First, the signals may be influenced by physiologic aspects across varying frequency ranges, such as Mayer waves. This will be mitigated through the application of moving average band-pass filters to raw signals, focusing only on the frequency range associated with cerebral vasomotion [31,32,60]. Second, dark skin/hair pigment is known to impair HbO/HHb measurements. The custom multi-channel NIRS system was selected as it only registers values if there is a true delta in optical densities for HbO/HHb, reducing the generation of erroneous values based on hyper-melanotic skin/hair, while the INVOS NIRS system states that dark skin pigment can cause poor performance. However, the INVOS system functions via a proprietary spatially resolved algorithm, in theory providing only true cerebral signal by scrubbing all scalp contamination and focusing on only reporting signals when there is a true physiologic variance in the signal. Access to the details of the algorithm is not possible as it is protected intellectual property. Third, with performing measurements using several sensors, an inherent issue of sensor cross-talk is possible. We will try to limit the cross-talk by ensuring proper distance is maintained between the sensors of the INVOS and OxyMon systems. We will leverage a minimum separation distance of 6 cm between both system’s optodes [61]. Fourth, scalp contamination of cerebral signals is a known issue, which will be accounted for by using a simultaneous short-channel optode setup (10 mm spacing), subtracting this signal from recording standard distance (30 mm) channels for a “pure” brain signal [62,63]. Contribution of neurovascular coupling [43,44,45,46] and vascular chemo-reactivity [64,65] will be accounted for through comprehensive testing. Local collaborations will facilitate needed expertise with neurovascular coupling with cognitive testing [66,67] and protocolized CO_2_ changes [42]. The adequacy of hyperemic response testing requires accurate CBF velocity signals through transcranial Doppler (TCD) [68,69], which will be facilitated using advanced robotic TCD systems (EMS-9D, Delica) obtained by the lab. This removes user-dependent error through hands-free insonation that can be co-applied with NIRS, as carried out in prior work [68,69]. Behavioural or psychiatric contributions to fNIRS’ signal variability will be mitigated by using only healthy volunteers for assessments [70,71,72,73,74]. Fifth, the proposed study protocol is limited to be conducted on healthy adults since some of the perturbation methods in the protocol could cause risk to neonatal/pediatric individuals. However, NIRS data are also relevant in neonatal/pediatric ICU’s to assess cerebral oxygenation [75,76], regional oxygenation [77], and CA [76]. Finally, the inter-disciplinary training of the assembled team, expertise with systems integration, signal analytics and advanced time-series methodologies ensure feasibility and success [17,20,33,55,56,58,69,78,79].

## 3. Procedure

### 3.1. Objective 1 Testing Protocol

The block design will consist of four blocks of different perturbations with a baseline rest between each block varying from 1 to 3 min depending on the time required to go back to normal levels after perturbation. The length of each block will vary depending on the number of repetitions along with the time required to go back to normal levels. The protocol is described in following subsections and is summarized in Table 2.

#### 3.1.1. Transient Hyperemic Response Test (6 min)

The transient hyperemic response will be tested using the carotid compression method. This test consists of five carotid compressions lasting 5 s each with a 60 s interval between each compression to allow normalization of the CBF to precompression levels [41], and the total time for this block is 6 min. This method is based on the compression of ipsilateral common carotid artery [41], and the response of the middle cerebral artery (MCA) blood flow velocity is assessed using a TCD probe (EMS-9D, Delica, Shenzen Delica Medical Equipment Co. Ltd, Shenzhen, China). The carotid compression is only accepted when no further decrease in blood flow velocity can be achieved, and stable conditions remain during the whole period of the compression. Otherwise, the compression is terminated and repeated again after 60 s. Satisfactory compression is typically considered to result in a reduction in systolic MCA velocity of 50%, at a minimum [80].

#### 3.1.2. Orthostatic Challenge Test (35 min)

The orthostatic challenge response is evaluated using lying-to-sit and sit-to-stand methods [81] five times where sitting and standing positions are held for 3 min each for a total block time of 35 min [82]. A baseline in lying position is collected at the start of the block; then, the position is changed from lying-to-sit for 3 min, and the position is then changed to standing for another 3 min.

#### 3.1.3. Vascular Chemo-Reactivity Test (12 min)

Vascular chemo-reactivity is assessed by varying CO_2_ concentrations through slow and fast breathing exercises with Capnostream 35 Portable Respiratory Monitor (Medtronic Canada) to monitor EtCO_2_ through nose clip. The breathing trials occur twice, and each of them consists of fast and slow trials along with an interval of normal breathing. The first time, the normal one, fast and slow breathing trials last 2.5 min, while the second time, these breathing trials last 1.5 min [42]. The normal respiratory rate for adults is 12 breaths/minute, the target slow respiratory rate is 5 breaths/minute and the fast respiratory rate is 25 breaths/minute as set by the metronome. It has been shown that the EtCO_2_ has a 10% increase during hypoventilation and almost 50% decrease during hyperventilation compared to normal breathing [42]. The breathing instructions will be communicated to the healthy subjects by a qualified lab personnel with the help of a metronome set at slow, normal or fast speeds (5, 12, or 25 breathes/min, respectively). This will be a great help to the volunteers to breathe in the correct frequencies themselves by listening to the set metronome rate. This block takes a total time of 12 min.

#### 3.1.4. Neurovascular Coupling Test (20 min)

Evaluate the neurovascular coupling using the automated neuropsychological assessment metrics (ANAM) general neuropsychological screening (GNS) clinical toolkit, which contains five tests explained as follows: (1) The standard continuous performance test assesses sustained attention, concentration and working memory by displaying individual characters in sequence and the volunteer responds only if the target letter shown previously is displayed. (2) The manikin test assesses 3D spatial rotation ability, left–right orientation, problem solving and attention by instructing the volunteer to indicate with which hand the man being displayed is holding the ball. (3) The pursuit tracking test measures the visuo-motor control by getting the volunteer to track a moving circle with a “+” inside where the pointer should remain inside the circle, as close to “+” as possible. (4) The switching test assesses divided attention, mental flexibility, and executive function by displaying both a manikin and a mathematical problem where an arrow indicates which problem to answer. (5) The Stroop test assesses the processing speed, selective attention, interference, and executive functioning with three trial blocks. The first block displays the words RED, GREEN or BLUE individually in black colour and the volunteer is instructed to say the word and enter a corresponding key. The second block displays XXXX’s in one of three colours where the volunteer says the color and then presses the corresponding key based on the colour. The third block has one of the three words displayed in a colour that does not match the name and the volunteer needs to press the key corresponding to the color. The neurovascular coupling block is expected to be completed in 20 min.

### 3.2. Objective 2 Testing Protocol

The testing design will consist of four phases of static and perturbation testing with a baseline rest of 2 min between perturbation phases due to the time required to go back to normal levels. The static phase will span 30 min and the three perturbation phases will take a total of half an hour. The protocol is described in the following subsections and is summarized in Table 3.

#### 3.2.1. Static Test (30 min)

Record the volunteers in a seated position for a total of 30 min in the static testing phase.

#### 3.2.2. Orthostatic Challenge Test (8 min)

Evaluate the orthostatic challenge response using the lying-to-sit and sit-to-stand methods [81] twice where sitting and standing positions are to be held for 2 min each for a total block time of 8 min [82]. A baseline in the lying position is collected at the start of the block; then, the position will be changed from lying-to-sit for 2 min, and the position will be then changed to standing for another 2 min.

#### 3.2.3. Vascular Chemo-Reactivity Test (9 min)

Evaluate the vascular chemo-reactivity by varying CO_2_ concentrations through slow and fast breathing exercises with a Capnostream 35 Portable Respiratory Monitor to monitor EtCO_2_ through nose clip. The breathing trials occur twice, and each of them will have fast and slow trials along with an interval of normal breathing. The first time, the normal one, fast and slow breathing trials last 2 min, while the second time, these breathing trials last 1 min [42]. As mentioned before, the normal respiratory rate for adults is 12 breaths/minute, and the target slow and fast respiratory rates as set by metronome will be 5 and 25 breaths/minute, respectively. Additionally, it has been shown in these slow and fast respiratory rates that the EtCO_2_ increases by 10% and decreases by almost 50%, respectively, as compared to normal breathing [42]. The breathing instructions will be communicated to the healthy subjects by a qualified lab personnel with the help of a metronome set at slow, normal or fast speeds (5, 12, or 25 breathes/min, respectively). This will be of great help to the volunteers by aiding them to breathe in the correct frequencies by listening to the set metronome rate. This block takes a total time of 9 min.

#### 3.2.4. Neurovascular Coupling Test (10 min)

The neurovascular coupling is evaluated using the ANAM Stroop test to assess the processing speed, selective attention, interference, and executive functioning with three trial blocks. The first block displays the words RED, GREEN or BLUE individually in black colour and the volunteer is instructed to say the word and enter a corresponding key. The second block displays XXXX’s in one of the three colours; the volunteer says the color and then presses the corresponding key based on the colour. The third block has one of the three words displayed in a colour that does not match the name, and the volunteer needs to press the key corresponding to the color. This testing phase is expected to be completed in 10 min.

## 4. Expected Results

We hope to verify the feasibility of the deriving CA maps of the entire convexity surface of the brain with high spatial and temporal resolutions using entirely non-invasive NIRS and niABP systems. Using the CA mapping system, we expect to show good variation and responsiveness in signals to the four perturbations (transient hyperemic response, orthostatic challenge, vascular chemo-reactivity, and neurovascular coupling). With the designed platform, we expect to explore the aging process and sex discrepancies in continuously assessed CA using the novel system statically. By calculating the slow-wave power of ABP and fNIRS waveforms using PSD, we expect to analyze the relationship between slow-wave power and the CA capacity in both cohorts. Furthermore, we expect to demonstrate CA map generation and point/interval predictions in real time using the system.

The software development of such an integrated CA mapping system is currently in its initial development stage, where CA heat maps can be produced in an “offline” mode with previously recorded CVR indices [30]. We expect to integrate the ability to capture real-time signal data and calculate CVR indices automatically in real-time to produce CA heat maps in “online” mode. Further, we expect to create prediction CA maps using time-series and machine learning forecasting techniques on adjustable time scales. Preliminary CA forecasting algorithms will be developed for univariate and multi-variate state models to demonstrate the feasibility of point and interval prediction of cerebral physiologic signals by deriving autoregressive integrative moving average (ARIMA) and vector ARIMA (VARIMA) [33,56].

Considering the short term, the development of this new wearable and portable imaging system will allow for the entirely non-invasive continuous assessment of regional differences in CA, improving our understanding of the impact of the aging process on vessel function, and potentially revolutionizing the way we monitor brain physiology in humans and other mammals. For the medium and long terms, success with this program carries the potential to advance the application of machine learning and computational approaches to the developed novel imaging platform, which can predict cerebral physiologic responses based on past data. In the long term, integration of the novel physiologic monitoring platforms with other big data from the proteome and genome of humans and other mammals will facilitate the expansion of our understanding of the fundamental mechanisms involved in cerebrovascular control.

## Figures and Tables

**Figure 1 mps-06-00058-f001:**
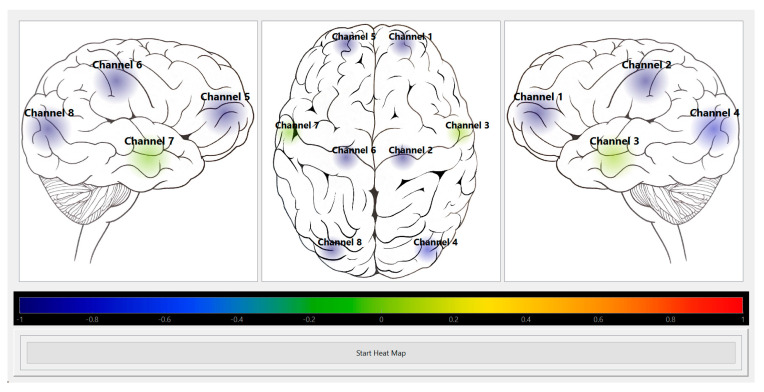
Heat map module visually displaying HbO autoregulation index for 8 channels. Shows a screenshot of the Python heat map module’s GUI running in “offline” mode for HbO autoregulation index of an 8-channel recording. The colour bar represents the CVR index scale from blue = −1 (intact CVR) to red = +1 (impaired CVR). *CVR*, *cerebrovascular reactivity*; *GUI*, *graphical user interface*; *HbO*, *oxyhemoglobin*. This figure is from Sainbhi et al. [30] and is used under Creative Commons Attribution License (CC BY) [40].

**Figure 2 mps-06-00058-f002:**
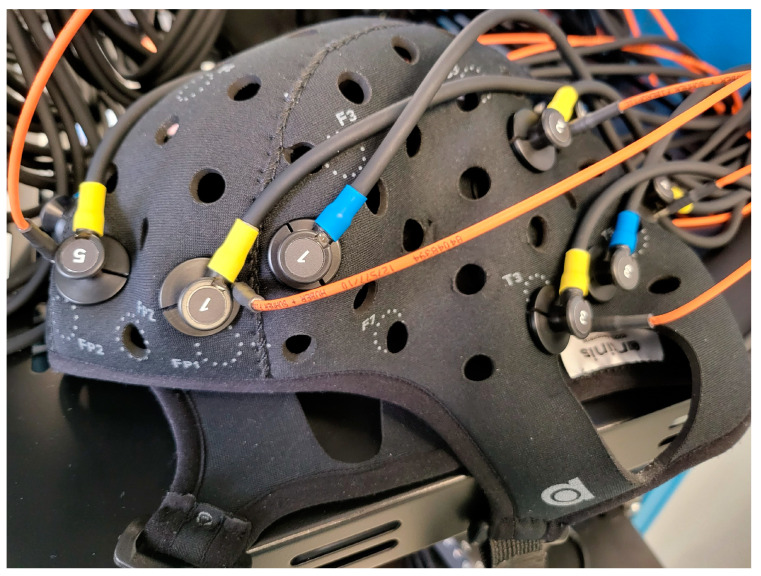
Displaying the OxyMon NIRS cap in close-up with transmitting optodes (in yellow) along with short receiver (in orange), and normal receiver (in blue) that create one short and one normal channel. *NIRS*, *near-infrared spectroscopy*. This figure is from Sainbhi et al. [30], is used under Creative Commons Attribution License (CC BY) [40], and is cropped from the original.

**Figure 3 mps-06-00058-f003:**
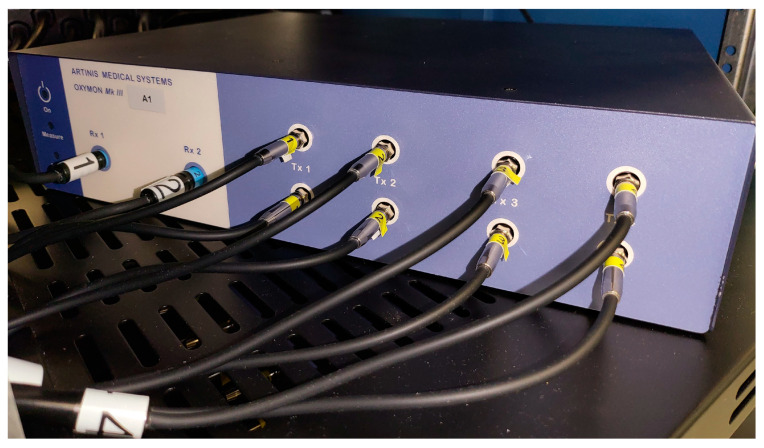
Displaying a portion of the multi-channel OxyMon Mk III device that is used to collect HbO, HHb, tHb, and HbDiff. *HbO*, *oxyhemoglobin*; *HHb*, *deoxyhemoglobin*, *tHb*, *total hemoglobin*; *HbDiff*, *difference between HbO and HHb*; *NIRS*, *near-infrared spectroscopy*. This figure is from Sainbhi et al. [30], is used under Creative Commons Attribution License (CC BY) [40], and is cropped from the original.

**Figure 4 mps-06-00058-f004:**
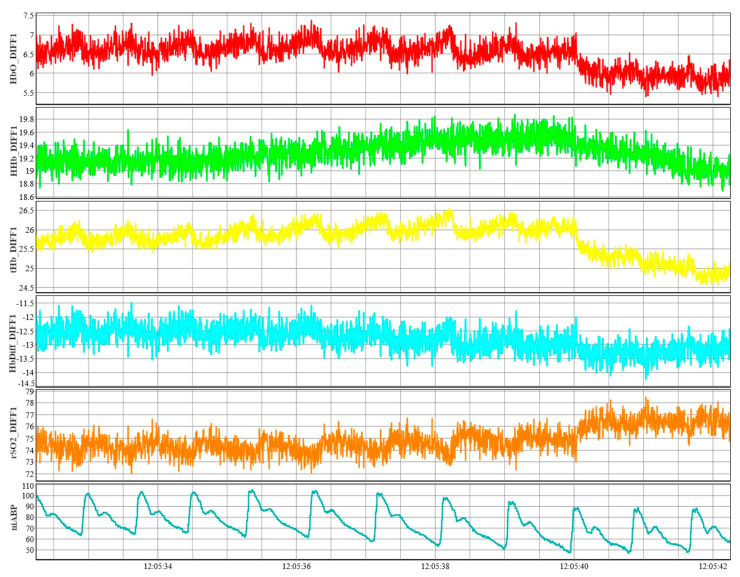
An example of NIRS signals of HbO, HHb, tHb, HbDiff, and rSO_2_ from channel 1 along with an example of niABP signal. The reference (short) channel NIRS signal have been subtracted from normal channel in the NIRS signals shown to eliminate scalp noise. *niABP*, *non-invasive continuous arterial blood pressure*; *DIFF1*, *differenced channel 1 signal as a result of subtracting short channel 1 signal from normal channel 1 signal*; *HbO*, *oxyhemoglobin*; *HHb*, *deoxyhemoglobin*, *tHb*, *total hemoglobin*; *HbDiff*, *difference between HbO and HHb*; *rSO_2_*, *regional cerebral oxygen saturation*. This figure is from Sainbhi et al. [30], is used under Creative Commons Attribution License (CC BY) [40], and is cropped from the original.

**Table 1 mps-06-00058-t001:** Derived CVR indices at each channel.

Abbreviation	CVR Metric	Correlation Between
HbOx	Oxyhemoglobin Index	HbO and ABP
HHbx	Deoxyhemoglobin Index	HHb and ABP
tHbx	Total Hemoglobin Index	tHb and ABP
HbDiffx	Hemoglobin Difference Index	HbDiff and ABP

ABP, arterial blood pressure; HbO, oxyhemoglobin; HHb, deoxyhemoglobin; tHb, total hemoglobin; HbDiff, difference between oxyhemoglobin and deoxyhemoglobin.

**Table 2 mps-06-00058-t002:** Objective 1 block testing phases.

Block	Test	Method	Perturbation Time	Interval Time	Repetition	Total Time
Rest Phase						
1	Transient hyperemic response	Carotid compression	5 s	1 min	5 times	~6 min
Rest Phase						
2	Orthostatic challenge	Lying-to-sit and sit-to-stand	Instantaneous	3 min	5 times	~35 min
Rest Phase						
3	Vascular chemo-reactivity	Breathing Exercises (fast/slow)	2.5 and 1.5 min	2.5 and 1.5 min	2 times	~12 min
Rest Phase						
4	Neurovascular coupling	ANAM GNS Clinical Toolkit	N/A	N/A	N/A	~20 min
Rest Phase						

ANAM, automated neuropsychological assessment metrics; GNS, general neuropsychological screening.

**Table 3 mps-06-00058-t003:** Objective 2 testing phases.

Block	Test	Method	Perturbation Time	Interval Time	Repetition	Total Time
1	Static	Normal sitting position	N/A	N/A	N/A	~30 min
2	Orthostatic challenge	Lying-to-sit and sit-to-stand	Instantaneous	2 min	2 times	~8 min
Rest Phase						
3	Vascular chemo-reactivity	Breathing Exercises (fast/slow)	2 and 1 min	2 and 1 min	2 times	~9 min
Rest Phase						
4	Neurovascular coupling	ANAM Stroop Test	N/A	N/A	N/A	~10 min

ANAM, automated neuropsychological assessment metrics; N/A, not applicable.

## Data Availability

Research ethics board approval at our institution does not facilitate free and open sharing of human data, regardless of the data being in a de-identified fashion. All such data are protected under both ethics and privacy acts within the Province of Manitoba, preventing such open sharing of data. All the data analyzed and used are available from the corresponding author upon reasonable request.

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
