# Peer review of "Non-Invasive Mapping of Cerebral Autoregulation Using Near-Infrared Spectroscopy: A Study Protocol"

_mps, 2023, doi:10.3390/mps6030058_

Round 1

Reviewer 1 Report

The study protocol is a well-written innovative study that will give an in-depth insight into the CA using a multi-channel NIRS system. The authors start with a ‘pilot’ study to study the feasibility of the monitoring set-up. Thereafter, they will perform a large study towards the relationship between age, sex, and the CA. I’ve several comments that could further improve the protocol.

 1.    The protocol of objective 2 starts with a thirty-minute baseline period of sitting to measure the CA as a “normative reference”. However, without induced slow oscillations, there might be a poor signal-to-noise ratio (DOI: 10.1152/japplphysiol.00853.2012). The authors might consider induced slow ABP-waves. In addition, I would recommend computing the power of the slow ABP-waves and presenting these results in the final study results.

2.    Line 43: “so CVR and CA are not entirely interchangeable since CVR can occur outside of the limits of autoregulation”. Please, insert reference.

3.    In the introduction the authors differentiate between CA and CVR. In the study only the CVR is assessed, but in the title is written “cerebral autoregulation mapping”. Throughout the text both terms are used. Please, clarify the text.

4.    Line 68: the refence [11] for the sentence “…provided in animal literature [11,12,19] seems incorrect as Ref 11 is a clinical study. Please, clarify.

5.    Line 62: the authors state that changes in rSO2 are a measure for volume changes. I would suggest to add that the rSO2 might also be a measure for changes in flow e.g. Brady et al. (2007) DOI: 10.1161/STROKEAHA.109.575159

6.    Line 151 – 155: The exclusion criteria are i.e., neurological, or cardiovascular disease. I would recommend listing more specifically the exclusion criteria in the protocol (e.g., color-blindness?)

7.    Line 418: The recruitment of healthy volunteers takes place from the local community/university. In addition, in line 152 is given that the authors intend to include at least 20 individuals per age decade. Please, comment how the authors will avoid a disbalance in age?

8.    Line 152-253: the authors state that the NIRS-system generates the true delta in optical densities for HbO/HHb, which reduces erroneous values for dark skin/hair pigment. What about the INVOS system?

9.    The rSO2 of the INVOS 5100c or 7100 system is used as “gold standard” (line 172). Did the authors consider collecting the optical density signals instead of the rSO2? (Melvin et al. 2022) DOI: 10.1016/j.bja.2021.10.035  

10.  The INVOS optodes seem to be located near two optodes of the Oxymon system. This might cause crosstalk between the Oxymon and INVOS signals. Please, comment on this.

11.  Line 190: the authors write a sampling frequency for rSO2 of ~1 Hz. The output of the INVOS-5100c has a sampling rate of ~0.2 Hz. Please, correct or clarify.

12.  Line 261 – 265: the authors seem to explain that they use the robot TCD to test the adequacy of the i.e., orthostatic test. Please, comment how this procedure is practically possible? Why not use TCD as “gold standard” for CA-assessment?

13.  In line 278 is the Carotid compression test described. Please, comment if ethical approval is obtained for this test, who will perform the test? Do the authors perform Carotid doppler studies prior to the measurements?

14.  Please, rephrase/clarify the headers of sections 3.1 and/or 3.2, as both objectives have the same header “testing protocol”

15.  Line 301-306: please clarify how the breathing instructions are communicated to the healthy subjects? It seems difficult to breathe in the correct frequencies yourself (e.g., audio instructions?)

16.  Table 2: please, write the full text for N/A in the legend

17.  Line 423 – 424: the appendix is not available for the reviewer

Reviewer 2 Report

Overall the methodological approach you propose seems sound.

However, while your call for fast data and the need multiple sites is interesting, I think you should be careful with the implication that what you suggest is the obvious way forward, and for this reason please add a comment on whether there is really clinical belief/interest for this approach, as many/most clinicians are not even convinced that even one site is worth trying.

One other clinical point you need to mention as a limitation to the method you propose is that NIRS brain monitoring of autoregulation and related data is relevant these days in neonatal care – not just adult ICUs re head trauma. Consequently, I doubt your methods for generating perturbation (e.g. carotid compression) would be acceptable  for use in sick newborns.

More importantly you need to provide more information where you talk about NIRS systems, as you actually only seem to be referring to CE or FDA clinically approved instruments like INVOS. Indeed these systems do measure only 1 or 2 sites - but some clinicians/researchers believe this is enough which you should mention - and are slow with a lot of averaging (up to 30 second filters), although the sampling rate itself is 1 Hz, or higher.

So please add more information to qualify your statement, and  distinguish between clinical and research systems. This is because research systems are available that are actually faster, and/or offer enough channels plus they are not too difficult to use. However, such systems are not CE or FDA  approved.

Do you have other alternative citations for your references 15-17?  – the full papers from these citations are not readily accessed – if not, you need to state in this paper what NIRS systems you used in the work you reference.

Reviewer 3 Report

Dear editor,

Authors proposed a study protocol of non-invasive mapping of cerebral autoregulation using near

infrared spectroscopy. This seems interesting and valuable and the paper is well organized. The paper is suitable published in this journal. However, this study requires a larger sample of human participants. In addition, it is necessary to provide a more comprehensive description of the feasibility of this study in the Introduction section.

Several comments:
Introduction

-Line 74 Please add more literature on the application of near-infrared spectroscopy technology in this field.

2. Experimental Design

-Line 99 As stated by the authors, this was a prospective observational study, so more research base should be added in the Introduction section to demonstrate the reliability of this study. The current Introduction is insufficient.

-Line 109 Provide a figure of NIRS used here or in 2.4. Physiologic Monitoring Systems section and describe some important parameters.

-Line 155 Please delete this sentence. It is not academic.

- If possible, please present some typical signals in 2.5. High-Frequency Data Acquisition section. Adding some figures to improve readability of the study protocol is encouraged.

-The authors used several statistical methods. Please describe some important statistical analysis methods in 2.7. General Statistical Methodologies.

Round 2

Reviewer 1 Report

The authors have satisfactorily responded to all my questions and updated the manuscript accordingly.

One minor comment that further improves the study protocol. I would recommend clarifying the legend of Figure 3, as the y-axis of rSO2% reads as 'dif_rSO2'. It is not clear in the legend which signal is being compared.

Reviewer 3 Report

The authors have addressed almost all of my comments. This paper can now be published in our journal.

Author Response

Thank you for your time in evaluating our submission.